# Functional Genes and Transcripts Indicate the Existent and Active Microbial Mercury-Methylating Community in Mangrove Intertidal Sediments of an Urbanized Bay

**DOI:** 10.3390/microorganisms12061245

**Published:** 2024-06-20

**Authors:** Guofang Feng, Sanqiang Gong

**Affiliations:** 1School of Environment and Energy, Peking University Shenzhen Graduate School, Shenzhen 518055, China; fenggf@pkusz.edu.cn; 2Shenzhen Key Lab of Industrial Water Saving & Municipal Sewage Reclamation Technology, Shenzhen Polytechnic University, Shenzhen 518055, China; 3Key Laboratory of Tropical Marine Bio-Resources and Ecology & Guangdong Provincial Key Laboratory of Applied Marine Biology, South China Sea Institute of Oceanology, Chinese Academy of Sciences, Guangzhou 510301, China

**Keywords:** mangrove intertidal sediment, mercury methylation, *hgcA*, transcript, correlation analysis

## Abstract

Mercury (Hg) methylation in mangrove sediments can result in the accumulation of neurotoxic methylmercury (MeHg). Identification of Hg methyltransferase gene *hgcA* provides the means to directly characterize the microbial Hg-methylating consortia in environments. Hitherto, the microbial Hg-methylating community in mangrove sediments was scarcely investigated. An effort to assess the diversity and abundance of *hgcA* genes and transcripts and link them to Hg and MeHg contents was made in the mangrove intertidal sediments along the urbanized Shenzhen Bay, China. The *hgcA* genes and transcripts associated with Thermodesulfobacteria [mainly Geobacteraceae, Syntrophorhabdaceae, Desulfobacterales, and Desulfarculales (these four lineages were previously classified into the Deltaproteobacteria taxon)], as well as Euryarchaeota (mainly Methanomicrobia and Theionarchaea) dominated the *hgcA*-harboring communities, while Chloroflexota, Nitrospirota, Planctomycetota, and Lentisphaerota-like *hgcA* sequences accounted for a small proportion. The *hgcA* genes appeared in greater abundance and diversity than their transcript counterparts in each sampling site. Correlation analysis demonstrated that the MeHg content rather than Hg content significantly correlated with the structure of the existent/active *hgcA*-harboring community and the abundance of *hgcA* genes/transcripts. These findings provide better insights into the microbial Hg methylation drivers in mangrove sediments, which could be helpful for understanding the MeHg biotransformation therein.

## 1. Introduction

The amount of mercury (Hg), a global pollutant, discharging from terrestrial to marine margins is (5500 ± 2700) Mg a^−1^, of which 72% is deposited in coastal sediments [1]. Therefore, marine–terrestrial ecotones have become the main reservoir of anthropogenic Hg pollutants. Mangrove sediments are ecologically important parts of coastal ecosystems, and mangrove sediments retain a considerable amount of the anthropogenic Hg reaching the marine ecosystem [2,3], which has become a paramount public health concern.

Hg can be converted into the neurotoxin, methylmercury (MeHg), via a microbially mediated methylation process in oxic and anoxic environments. Hg methylation has raised concerns because MeHg poses extreme threats to ecological security and human health [4]. Prior studies concerning the physiochemical analysis of the total Hg (THg) and MeHg contents and transformations have confirmed the occurrence of MeHg formation and bioaccumulation in mangrove sediments [5,6], which may have a nonnegligible impact in global Hg cycling. One study has implicated sulfate-reducing bacteria (SRB) as important Hg methylators in mangrove sediments, whereas other microbes such as iron-reducing bacteria (IRB) and methanogenic archaea (MGA) may also contribute to MeHg formation therein [7]. However, such research is not enough to identify the phylogenetic members responsible for Hg methylation in mangrove sediments. Specific knowledge of the Hg methylators would provide valuable information on their physiologies and ecologies, thereby providing additional insight into the specific controls on Hg methylation in this ecosystem.

The identification of Hg methyltransferase gene *hgcA* has untangled the genetic basis for microbial Hg methylation and provided the methods to directly characterize the microorganisms involved in Hg methylation in various habitats [8]. In addition, the *hgcA* phylogeny of known Hg-methylating strains tends to track 16S rRNA gene phylogeny well [9]. Therefore, based on *hgcA*-targeted or metagenome-wide molecular analysis, taxonomic information of microbial members with Hg methylation potential has been uncovered from various ecosystems. For example, *hgcA*-based sequencing analysis has revealed the putative Hg-methylating communities dominated by Syntrophobacterales SRB, Geobacteraceae IRB, Methanomicrobia MGA, or Ruminococcaceae in different habitats, such as the Everglades soils [10], wetland soils [11], paddy soils [12], water-level fluctuating zones [13], plateau lake sediments [14], boreal lake sediments [15], or boreal forest soils [16]. An in vivo *hgcA* analysis of publicly available microbial genetic information predicted a wider spread of anaerobic microbes than previously thought, with Hg methylation potentially inhabiting a broad range of niches [17].

In addition, one metagenomic study revealed abundant atypical Hg methylators, such as Aminicenantes, Kiritimatiellaeota, and Spirochaetes, in sulfate-impacted lake sediments [18]. Another metagenomic study even identified the aerobic *Nitrospina* contributing to Hg methylation in Antarctic ice [19], and as the dominant and widespread key MeHg producers in the oxic subsurface waters of the global ocean [20]. Thus, *hgcA*-based revelation is useful to identify the members of the Hg-methylating community from various ecosystems, which has complementary advantages to previous MeHg measurement studies.

Noteworthily, most current molecular analyses are performed to predict the existence of putative Hg-methylating microorganisms in environments via *hgcA* gene-based surveys. However, such studies cannot differentiate metabolically active cells from dormant ones in complex ecosystems, which perhaps misinterpret the real Hg methylation functioning community therein. As a comparison, functional transcription is conducting metabolic processes and is sensitive to the environmental fluctuations [21]. Accordingly, transcript-based studies can reflect the metabolic activities of the microbial community in the environment [22,23]. For example, *hgcA* transcript-based studies have revealed the metabolic activity of the Hg-methylating community in a landfill [24] and the river downstream of a chloralkali plant [25]. However, compared with the reported *hgcA* gene-based studies, descriptions of the active Hg-methylating community in environments, such as mangrove sediments, are scarce.

As a hotspot of Hg methylation [26], Hg speciation, bioaccumulation, and transportation in mangrove sediments might be related to the local contamination status, economic development, and geomorphic and hydrological conditions [27]. As an example, Shenzhen Mangrove Nature Reserve, located in the northeastern coast of Shenzhen Bay, is a typical mangrove area in close proximity to a heavily populated and urbanized industrial city [28]. This mangrove area has been proven to play an important role in the retention of many anthropogenic wastes, including Hg pollutants [29,30,31]. Previous studies showed that the Hg content in the mangrove intertidal sediments (154.7–218.4 ng g^−1^) of Shenzhen Bay surpassed the background level (71.0 ng g^−1^) [32]. Also, the MeHg content in Shenzhen Bay sediments (0.85–1.57 ng g^−1^) was higher than in most other estuaries worldwide [6]. Therefore, the Shenzhen mangrove sediments deserve microbial Hg methylation surveys.

Hitherto, our understanding of the Hg methylation process in mangrove ecosystems is constrained by a lack of in-depth studies on the Hg methylation potential of the corresponding microbial consortia in this ecologically important environment. A detailed characterization of the structure and activity of the microbial Hg-methylating community, and their correlation with THg and MeHg contents, will be important to guide research for monitoring and eliminating MeHg hazards in mangrove ecosystems. Given the above, the objectives of this study were to assess the phylogenetic diversity and transcriptional activity of microbial consortia driving Hg methylation in mangrove sediments. Moreover, the correlation between THg/MeHg contents and Hg methylating genes/transcripts was assessed concurrently.

## 2. Materials and Methods

### 2.1. Sediment Sampling and Processing

Soil samples were collected from the mangrove intertidal zone located in the Shenzhen Mangrove Natural Reserve in Shenzhen, China, on 15 May 2019. Surface sediment samples (0–5 cm) from three sites were collected following the previous strategy [33], namely site M1 (114°0′14″ E, 22°31′31″ N), M2 (114°0′23″ E, 22°31′30″ N), and M3 (114°0′34″ E, 22°31′28″ N) (Figure 1). Triplicates of the samples from each site were transferred into 10 volumes of RNAprotect Bacteria Reagent (Qiagen, Hilden, Germany) and stored at −80 °C before total DNA or RNA extraction. The other triplicates of the samples from each site were immediately frozen by CO_2_ dry ice after being collected for the analysis of THg and MeHg contents in the laboratory.

### 2.2. Physicochemical Parameters, THg, and MeHg Content Analysis

Total organic carbon (TOC), and total nitrogen (TN) contents in sediment samples were measured with a TruMac CNS Macro Analyzer (LECO, St. Joseph, MI, USA). The pH and redox potential (*Eh*) values of each sample were determined using a pH/ORP/Temp Portable Meter 6010N (JENCO, San Bernardino, CA, USA) prior to collection.

The pretreatment and measurement of the THg and MeHg contents in the sediment samples were performed according to the previous procedure [34]. After preparation, the THg and MeHg contents in the sediment were measured using the ZYG-II cold vapor atomic fluorescence spectrometer (CVAFS) (Daji Electric Instrument, Hangzhou, China). Briefly, sediment samples for THg measurement were digested with a mixture of HNO_3_ and HCl at 105 °C for 2 h, and then BrCl was added to oxidize all forms of Hg to Hg^II^. After filtering the solution, SnCl_2_ was added to reduce Hg^II^ to Hg^0^. Volatilized Hg was then concentrated on a gold trap heated to release mercury vapors, which were detected with a CVAFS system according to USEPA Method 1631 [35]. The samples analyzed for MeHg were treated with KBr + CuSO_4_/solvent and KOH–methanol/solvent extraction, and the MeHg levels were determined using the CVAFS system following USEPA Method 1630 [36].

### 2.3. DNA, RNA Extraction, and cDNA Synthesis

RNA protector-fixed samples were melted on ice, centrifuged at 10,000× *g*, 4 °C for 10 min, and the supernatant was discarded. DNA or RNA was extracted in triplicate from 0.2 g of pellets with the PowerSoil DNA isolation kit (MoBio, Carlsbad, CA, USA) or the PowerSoil Total RNA Isolation Kit (MoBio) following the manufacturer’s instructions. RNase-free DNase I (Fermentas, Hanover, PA, USA) was used to digest the potential residual DNA in the total RNA at 37 °C for 60 min. RNA quality and integrity were checked by gel electrophoresis and by examining the A260/A280 ratio (1.9–2.0) using a NanoDrop spectrophotometer (Thermo Scientific, Waltham, MA, USA). The final RNA concentration and purity were quantified using the Qubit system (Invitrogen, Darmstadt, Germany).

First-strand cDNA synthesis was performed using the SuperScript First-Strand Synthesis System (Invitrogen, Carlsbad, CA, USA). Each reaction volume was 10 µL containing 100 ng RNA, 0.5 µL random hexamers primer (50 ng µL^−1^), 5 µL cDNA Synthesis Mix, and RNase-free water. This reaction was carried out by incubating at 25 °C for 10 min, then at 50 °C for 50 min, and terminating at 85 °C for 5 min. All the cDNA aliquots were stored at −80 °C before polymerase chain reaction (PCR) amplification.

### 2.4. PCR, Cloning, and Sequence Analysis

The *hgcA* sequences were amplified from DNA or cDNA templates using previously published *hgcA* primer pair hgcA261F (5′-CGGCATCAAYGTCTGGTGYGC-3′)/hgcA912R (5′-GGTGTAGGGGGTGCAGCCSGTRWARKT-3′) [15], which located the 249~269 and 877~903 regions of the *hgcA* gene (B2D07_07835) from the methylating species *Desulfococcus multivorans* DSM 2059. The forward primer, hgcA261F,  targets the highly conserved putative cap helix, while the reverse primer, hgcA912R,  targets a less-conserved motif between the third and fourth predicted transmembrane helices near the C-terminus, producing an amplicon of ~650 bp [8]. PCR was conducted in a total volume of 40 μL containing 10 ng DNA or 2 μL cDNA (equivalent to 4 ng RNA), 0.4 μL of each primer (10 μM solution), and 20 μL TaqMasterMix (CoWin Biotech, Beijing, China). PCR amplification was performed on a Thermocycler (Eppendorf, Hamburg, Germany). The optimized PCR cycling parameters were the following: 95 °C for 5 min, followed by 30 cycles of 95 °C for 40 s, 60 °C for 30 s, 72 °C for 1 min, and a final extension at 72 °C for 10 min. For negative control, a similar procedure was carried out using purified RNA in place of cDNA to ensure that there was no genomic DNA contamination. DNA or cDNAs of triplicates for each sampling site were PCR-amplified. The presence and size of the amplification products were estimated by gel electrophoresis (1.5% agarose gel). The PCR products were purified with MinElute Gel Extraction Kit (Qiagen). Then, the PCR products were cloned with pUCm-T Vector Rapid Cloning Kit (Sangon Biotech, Shanghai, China) and transformed into *Escherichia coli* DH5*α* competent cells (CoWin Biotech) according to the standardized instructions. The positive clones were randomly selected in a Luria-Bertani agar plate containing ampicillin (50 μg mL^−1^) and identified by PCR screening with vector-specific M13 primers. Over 100 clones from each clone library were sequenced on an ABI 3100 capillary sequencer (Generay Biotech, Shanghai, China).

### 2.5. Clone Library Construction and Diversity Assessment

In total, three *hgcA* gene-based clone libraries from DNA templates (M1D, M2D, and M3D) and three *hgcA* transcript-based libraries from RNA templates (M1R, M2R, and M3R) corresponding to the sites M1, M2, and M3 were constructed. For each library, the sequenced clones were trimmed manually using ClustalW implemented in MEGA X with default settings [37]. Sequences from clone libraries were clustered into *hgcA* phylotypes defined at a 20% nucleotide dissimilarity threshold [12], using the Mothur package [38]. Sequences

from bacterial amoA clone

libraries (284) were clustered into

OTUs defined at 3% nucleotide

sequence dissimilarity

The Sobs (number of the observed phylotypes), Shannon, and InvSimpson diversity indices and Chao and Ace richness estimators were calculated using the Mothur package to reflect the richness and diversity of *hgcA* genes or transcript sequences. Rarefaction curves and coverage estimators were determined using the Mothur package to estimate whether the sequencing depth was enough to cover most of the *hgcA* genes or transcripts in each library at the 20% nucleotide dissimilarity threshold. A Venn diagram illustrating the shared similarity of the *hgcA* phylotypes between different libraries was generated using the nVenn software (v1.0) [39].

### 2.6. Phylogenetic Analysis

To construct the phylogenetic tree, one representative sequence from each *hgcA* phylotype and its closest sequence retrieved from the NCBI Nucleotide Database were aligned using ClustalW implemented in MEGA X [37]. The evolution models of the sequence collections were evaluated using Akaike information criteria tests implemented by jModelTest [40]. The maximum-likelihood tree was constructed in MEGA X based on the best-fit evolution model with a gamma shape parameter derived from jModelTest. Bootstrap analysis was used to estimate the reliability of the phylogenetic reconstructions (with 1000 reassemblages). Taxonomic classifications can be assigned with the lowest common ancestor (LCA) of the subtree where the sequence was placed by the ‘guppy classify’ command [41] with a classification cutoff of 80%.

### 2.7. Quantification Assessment

Quantitative PCR (qPCR) for enumeration of *hgcA* genes or transcripts was performed using the primer pair hgcA261F/hgcA912R, following the previous categorization strategy [25]. In detail, PCR was performed in a total volume of 25 μL containing 12.5 μL of SYBR Premix Ex Taq II (Takara, Dalian, China), 1 μL of DNA or cDNA template (tenfold serial dilution), and 0.5 μL each of 10 μM solutions of primers. The PCR thermocycling steps were set as follows: 95 °C for 5 min and 40 cycles at 95 °C for 45 s, 60 °C for 45 s, and 72 °C for 45 s. A similar procedure was carried out using purified RNA in place of cDNA to ensure that there was no genomic DNA contamination. A standard curve (log-linear R^2^ > 0.99) was generated by plotting the relative fluorescent units at a threshold fluorescence value (*C_T_*) versus the logarithm of the copy number of the constructed pUCm plasmids containing a 606-bp *hgcA* gene fragment (GenBank ID MN475572) in a dilution series that spanned from 10^1^ to 10^7^ gene copies per reaction. All standard dilutions were prepared in 10 ng μL^−1^ aqueous tRNA solution (Sigma-Aldrich, Steinheim, Germany). Plasmid DNA was extracted using the PurePlasmid Mini Kit (CoWin Biotech), and the plasmid content was measured using the Qubit system (Invitrogen). Since the lengths of the vector and PCR insert are known, the copy numbers of the *hgcA* gene sequences were directly calculated according to the reported formula: copy numbers μL^−1^ = (A × 6.022 × 10^23^) × (660 × B)^−1^, where A is the plasmid content (g μL^−1^), B is the length (bp) of the recombinant plasmid pUCm-*hgcA*, 6.022 × 10^23^ is the Avogadro’s number, and 660 is the average molecular weight of 1 bp [42]. The PCR efficiency (*E*) of 94% was calculated from the slope of the standard curve by using the formula *E* = 10^[−1/slope]^ [43]. After the qPCR assay, the specificity of amplification was verified by the generation of melting curves (in steps of 0.5 °C for 5 s, with temperatures ranging from 60 to 95 °C), and the qPCR product size was checked by 1% agarose gel electrophoresis.

### 2.8. Statistical Analysis

A Student’s *t*-test was conducted using the IBM SPSS Statistics 19.0 software with statistical significance below 0.05 (*p* < 0.05). The Bray–Curtis matrix distance and thetaYC matrix distance, based on the *hgcA* phylotypes identified in all the DNA and cDNA libraires, were calculated via the corresponding Mothur commands. The Bray–Curtis distance was visualized by hierarchical clustering heatmap analysis, and thetaYC distance was by principal coordinate analysis (PCoA). Comparisons between the *hgcA*-harboring community from different sites (M1, M2, M3) at different levels (gene, transcript) were analyzed using an analysis of similarity statistics (ANOSIM) on the Bray–Curtis and thetaYC indices through Mothur. Canonical correlation analysis (CCA) was performed to analyze the correlation between the *hgcA*-harboring community structure and THg or MeHg content using Canoco 5. Pearson’s correlation coefficient (r) was conducted to estimate the correlation between THg or MeHg content and *hgcA* gene or transcript abundance.

## 3. Results

### 3.1. Physicochemical Characteristics, THg, and MeHg Content in Mangrove Sediments

A physicochemical parameters analysis showed a weakly acidic (pH 6.67~6.83) and reduced (*Eh* −121~−107 mV) environmental background (Appendix A). Various THg and MeHg contents (ng g^−1^ soil) were revealed in the sampling sites. The THg content (ng g^−1^ soil) in M1, M2, and M3 were 160.2 ± 4.8, 157.1 ± 1.5, and 157.4 ± 4.1, respectively, which were not significantly different from each other (*p* > 0.05), whereas the MeHg content (ng g^−1^ soil) in M1, M2, and M3 was 1.53 ± 0.13, 0.86 ± 0.12, 1.22 ± 0.11, respectively, which were obviously different from each other (*p* < 0.05) (Figure 2A). Thus, the MeHg contents were not consistent with the Hg contents in the sampling sites of mangrove sediments.

### 3.2. Diversity of hgcA Sequences in Mangrove Sediments

The sequence diversity was analyzed on the *hgcA* sequences. A total of 648 *hgcA* sequences (308 gene sequences and 340 transcript sequences) were obtained from sites M1, M2, and M3 and fell into 118 phylotypes at a 20% nucleotide sequence cutoff. The Ace and Chao richness estimators and Shannon and InvSimpson diversity indices predicted more *hgcA* phylotypes and higher richness and diversity of *hgcA* sequences in the DNA library than their corresponding RNA library in each sampling site (Table 1). The coverage statistic showed that 67~80% of *hgcA* sequences in each clone library were included (Table 1), indicating that more sequences would contribute higher richness and diversity of *hgcA* sequences in the sampling sediments. These coverage values were consistent with the rarefaction curve analysis since none of the rarefaction curves reached an asymptote (Figure 2B). Venn analysis showed that most of the phylotypes (75 of 118) were shared between the DNA clone library and their corresponding RNA clone library in each site, whereas a few phylotypes (3 of 118) were shared among all three sites (Figure 2C). Thus, the *hgcA* phylotype composition might be different among the sampling sites in the investigated mangrove sediments.

### 3.3. Dissimilarity of Different hgcA–Harboring Communities

PCoA based on the thetaYC matrix distance showed that the *hgcA*–harboring communities from three sites were clearly separated by axes PCoA1 (explaining 48.8% of the variation) and PCoA2 (explaining 39.6% of the variation) (Figure 3A). However, the gene–based library cannot be separated from its corresponding transcript–based library in each site (Figure 2A). This revelation was consistent with the hierarchical clustering heatmap analysis based on the Bray–Curtis distances between Hg–methylating communities (Figure 3B). The ANOSIM results based on Bray–Curtis or thetaYC indices showed that the dissimilarity of the *hgcA*–harboring community structure between different sampling sites was significant (*p* < 0.05). Thus, the *hgcA*–containing community may vary within different sites of mangrove sediments.

### 3.4. Phylogenetic Information of hgcA Phylotypes

A phylogenetic analysis showed that all 118 *hgcA* phylotypes fell into 13 distinct clusters. These clusters included five Thermodesulfobacteria clusters [i.e., Geobacteraceae, Syntrophorhabdaceae, Desulfobacterales, and Desulfarculales (these four taxa were previously classified as Deltaproteobacteria [44]), and Unclassified I cluster], three Euryarchaeota clusters (i.e., Methanomicrobia, Theionarchaea, and Unclassified II cluster), one Chloroflexota cluster, one Nitrospirota cluster, one Lentisphaerota cluster, one Planctomycetota cluster, and the unclassified III cluster (Figure 4).

Among the Thermodesulfobacteria clusters, *hgcA* phylotypes falling into the Geobacteraceae cluster were affiliated with some IRB species, including many putative Hg methylators, e.g., *Geobacter* spp. (ABQ24695, WP_145023549, WP_012647565, WP_010942086). The Syntrophorhabdaceae cluster contained three *hgcA* phylotypes clustered with *Syntrophorhabdus* spp. (OPX92424, OPY73706). In addition, the *hgcA* phylotypes in Desulfobacterales and Desulfarculales clusters were loosely related to some putative Hg–methylating SRB species, e.g., *Desulfospira joergensenii* (WP_153307583), *Desulfococcus multivorans* (AHN16617), *Desulfofaba hansenii* (WP_100393491), *Desulfocarbo indianensis* (KMY66723), and *Desulfarculus* sp. (RJX35066). The Methanomicrobia cluster contained 38 *hgcA* phylotypes that were related to some putative Hg–methylating MGA species, e.g., *Methanospirillum hungatei* (WP_143709353), and *Methanolobus tindarius* (WP_023846500). The Theionarchaea cluster contained five phylotypes, which were related to Theionarchaea archaeon (KYK30264). The Chloroflexota cluster comprising six phylotypes was loosely related to *Dehalococcoides mccartyi* (WP_148284408) and the uncultured sequences from alder swamp soils, lake sediments, and paddy soils. The Nitrospirota cluster contained two phylotypes distantly related to the Nitrospirota bacteria in estuary sediment [45]. The Lentisphaerota cluster and Planctomycetota cluster only contained one phylotype, respectively. In addition, 13 phylotypes falling into the Unclassified III cluster cannot be classified into any definite lineage and are related to the uncultured sequences derived from paddy soils, alder swamp soils, lake sediments, estuarine sediments, and freshwater wetland soils [10,12,18,46]. Among the 118 phylotypes, there were 37 phylotypes accounting for >0.5% of the total *hgcA* sequences; these phylotypes accounted for 70.5~79.1% of the *hgcA* sequences in each clone library, representing the dominant *hgcA* phylotypes in the sampling sites, whereas the remaining 81 *hgcA* phylotypes represented 20.9~29.5% of the *hgcA* sequences in the corresponding library, contributing to the richness and diversity of *hgcA* phylotypes in mangrove sediments. A heatmap analysis of these *hgcA* phylotypes showed that the top three phylotypes, i.e., Methanomicrobia OTU001 (MN475604), Geobacteraceae OTU002 (MN475677), and Desulfobacterales OTU003 (MN475657), were the only three phylotypes shared among the six libraries, as revealed by the Venn analysis (Figure 2C), and these three phylotypes alone accounted for 4.1~22.8% of the clones in each clone library (Appendix A).

### 3.5. CCA between the THg and MeHg Contents and hgcA–Harboring Community

Correlations between the structure of *hgcA* genes or the transcript–harboring community and THg or MeHg contents were analyzed via CCA (Figure 5A). The two CCA dimensions explained 92.3% of the cumulative variance in the *hgcA*–harboring community–THg/MeHg content relationship. THg or MeHg content has a similar influence on the *hgcA* genes and transcript–harboring community in each sampling site (Figure 5A). However, the influence of THg or MeHg content on the *hgcA*–harboring community in various sampling sites was different from each other. A Monte Carlo permutation test showed that, the MeHg content significantly affected the structure of the *hgcA* gene/transcript–harboring community in different sampling sites of mangrove sediments (F = 3.5, *p* < 0.05), whereas the THg content was not statistically significant (F = 1.2, *p* > 0.05).

### 3.6. Abundance of hgcA Genes and Transcripts in Mangrove Sediments

The *hgcA* gene and transcript abundances in sediments were quantified using qPCR assessment in the M1, M2 and M3 sites. As shown in Figure 5B, *hgcA* gene abundance in M2 [(8.35 ± 0.57) × 10^7^ g^−1^ soil] > M1 [(5.83 ± 1.28) × 10^7^ g^−1^ soil] and M3 [(4.48 ± 0.52) × 10^7^ g^−1^ soil] (*p* < 0.05); whereas *hgcA* transcript abundance in M2 [(1.95 ± 0.16) × 10^7^ g^−1^ soil] > M1 [(7.88 ± 1.32) × 10^6^ g^−1^ soil] > M3 [(3.02 ± 0.49) × 10^6^ g^−1^ soil] (*p* < 0.05) (Figure 5B). In each site, the gene abundance was significantly higher than its transcript counterpart abundance (*p* < 0.05). Moreover, the number of transcripts per gene, determined as the transcript–to–gene ratio, was measured as a transcriptional activity metric of *hgcA* genes in mangrove sediments. The mean transcript–to–gene ratio of *hgcA* in M1, M2, and M3 were 0.135, 0.198, and 0.067, respectively. Therefore, the abundance and transcriptional activity of *hgcA* genes in the investigated mangrove sediments differed between sampling sites.

### 3.7. Pearson’s Correlation between THg or MeHg Content and hgcA Gene or Transcript

Pearson’s correlation analysis showed that THg content was not significantly correlated with the abundance of *hgcA* genes or transcripts (r = 0.352 and 0.364, *p* > 0.05) (Table 2), whereas a positive linear correlation between MeHg content and the abundance of *hgcA* genes or transcripts (r = 0.997 and 0.999, *p* < 0.05) was identified from the sampling sites of mangrove sediments (Table 2). In addition, neither THg content nor MeHg content was significantly correlated with the *hgcA* transcript–to–gene ratio (r = 0.527 and 0.987, *p* > 0.05) (Table 2). This finding highlighted that the MeHg content was remarkably correlated to the *hgcA* gene or transcript abundance in the investigated sites of mangrove sediments.

## 4. Discussion

As one of the hotspots of Hg accumulation in aquatic environments [2], relatively high Hg methylation rates have been confirmed in mangrove intertidal sediments [5]. Here, diverse putative Hg–methylating phylotypes with transcriptional activity were uncovered, which provided a snapshot of the active Hg–methylating population in mangrove sediments and other environments [25,47,48]. 

The MeHg contents in the sampling sites of mangrove intertidal sediments showed obvious heterogeneity (Figure 1), in agreement with the previous measurement of the MeHg contents in this area [6]. This may be due to the fact that soils in mangrove sediments were highly localized and heterogeneous [49]. Hg content transformations in marine sediments were directly affected by natural and anthropogenic factors, including the variety of Hg species, terrestrial pollutants (sulfate, organic matter, ammonium), chemical and physical geomorphic properties (pH, silt, and clay fractions), and indirectly affected by economic status [50], which may contribute to the heterogeneity of MeHg contents therein. The MeHg content was then a significant factor affecting the *hgcA*–harboring community structure. It has been proposed that MeHg could be exported from metabolically methylating cells to reduce the toxicity to microbial methylators [51]. Such a finding was similar to that in paddy soils anthropogenically treated with different mercury species [52], but different from that in paddy soils adjacent to mercury mining areas, where MeHg content was not the significant factor affecting the Hg–methylating community structure [12,53], indicating that other environmental factors may significantly affect the Hg–methylating community.

The richness, diversity, and quantitative abundance of *hgcA* genes were higher than that of their transcript counterparts in each sampling site. This finding indicated the apparent redundancy of *hgcA* genes in mangrove sediments. Thus, only a fraction of the *hgcA*–harboring microorganisms may be metabolically active, while the remaining ones might be in their dormant period or an extremely low activity state below the PCR sensitivity threshold. Similar conditions have been found for nitrifying microbes in marine sponges [54] and for denitrifying bacteria in oligotrophic groundwater [55]. Previous studies concluded that functional gene redundancy is a fundamental property of microbial communities, which could enhance their metabolic plasticity and shape their response to environmental forcing [56,57].

Phylogenetic analysis demonstrated that all the *hgcA* phylotypes fell into 13 clusters, indicating the complexity of the putative Hg–methylating community in mangrove sediments. The three most abundant *hgcA* phylotypes were shared among the six libraries and associated with Methanomicrobia MGA, Geobacteraceae IRB, and Desulfobacterales SRB, respectively. This revelation agreed with a previous metabolic inhibitor treatment report, which revealed that SRB were important Hg methylators in mangrove sediments, while the participation of IRB and MGA in Hg methylation should not be neglected therein [7]. In addition, a large fraction of bacterial *hgcA* sequences fall into the Geobacteraceae, Syntrophorhabdaceae, Desulfobacterales, and Desulfarculales clusters of the Thermodesulfobacteria phylum. Hg–methylating Thermodesulfobacteria (previously classified as Deltaproteobacteria) have been uncovered from various habitats and account for a high relative abundance of total *hgcA* sequences, such as in boreal forest soils [16], paddy soils [53], lake sediments [15], mountain glaciers [58], and coastal canyon sediments [59]. Also, a few fractions of bacterial *hgcA* phylotypes are gathered within the Chloroflexota, Nitrospirota, Planctomycetota, and Lentisphaerota clusters. The *hgcA* sequences associated with these clusters were also recently uncovered with a low relative abundance in various ecosystems, such as salt marsh and peat soils [60], thawing permafrost [61], mountain glaciers [58], coastal canyon sediments [59], eutrophic lake sediments [62], suboxic seawaters [48], marine particles [63], and creek sediments [60]. Additionally, most of the archaeal *hgcA* sequences were gathered within the Methanomicrobia cluster. Former studies indicated that this lineage was a typical and dominant archaeal Hg–methylating lineage in sawgrass marsh, alder swamp [11], water–level fluctuating zone soils [13], and sulfate–impacted lakes [18]. In addition, a few archaeal *hgcA* sequences were related to Theionarchaea. Theionarchaea *hgcA* sequences were once identified in an in vivo screening of reported metagenomes [64] and now were confirmed with transcriptional activity in mangrove sediments. This finding indicated that, besides Methanomicrobia, Theionarchaea might be a useful archaeal Hg–methylating population in the investigated mangrove sediments. Noteworthily, over 40 *hgcA* phylotypes did not show definitive links to the known Hg–methylating microbes, indicating that some novel yet unidentified Hg–methylating species inhabit the mangrove sediments and contribute to Hg methylation therein. Taken together, a complex and diverse microbial community may drive Hg methylation in mangrove sediments. Moreover, since some of the *hgcA* phylotypes cannot be associated with any definite lineage, further work is necessary to define the unclassified Hg methylators and better understand their contributions in mangrove sediments.

The quantitative abundance of *hgcA* genes and transcripts was also detected in this study. The observed MeHg content exhibited an obvious positive correlation with *hgcA* gene and transcript abundance in mangrove sediments, which was consistent with the findings in Hg mining area paddy soil [12], the water–level fluctuating zone [13], and paddy soils [65]. As a comparison, the THg content was not related with the *hgcA* gene or transcript abundance. This could be because the THg was composed of different Hg species that may have divergent effects on the Hg bioavailability and microbial methylation capacity [52]. Therefore, our results suggest that direct detection of *hgcA* genes/transcripts would be an available method to assess the MeHg accumulation potential in the investigated mangrove sediments. Nevertheless, MeHg accumulation may be not only significantly associated with the *hgcA* gene and transcript abundances but also with some other biotic and abiotic factors, such as wind fluctuation interactions [66] and MeHg–demethylating bacteria dynamics [67]. Consequently, more works are needed to confirm the factors that affect microbially mediated MeHg production and accumulation in mangrove sediments as well as other environments.

## 5. Conclusions

The structure characteristics and transcriptional activity of Hg–methylating communities in mangrove sediments were revealed by *hgcA*–based analysis. A complex and diverse community was uncovered, spanning the Thermodesulfobacteria (mainly Geobacteraceae and Desulfobacterales), Euryarchaeota (mainly Methanomicrobia), Chloroflexota, Nitrospirota, Planctomycetota, and Lentisphaerota taxa. The structure and *hgcA* abundance of the existent and active putative Hg–methylating communities varied among different sampling sites, with *hgcA* genes displaying greater diversity and abundance than their transcript counterparts in each site. The MeHg content significantly affected the structure of the existent/active putative Hg–methylating community and obviously correlated with *hgcA* gene/transcript abundance. Clarification of the microbial Hg–methylating communities and their correlations with THg and MeHg contents would confirm the microbial drivers in Hg methylation and help minimize Hg methylation in mangrove sediments. More efforts are needed to gain better insights into the contributions of microbial communities involved in Hg methylation in mangrove sediments.

## Figures and Tables

**Figure 1 microorganisms-12-01245-f001:**
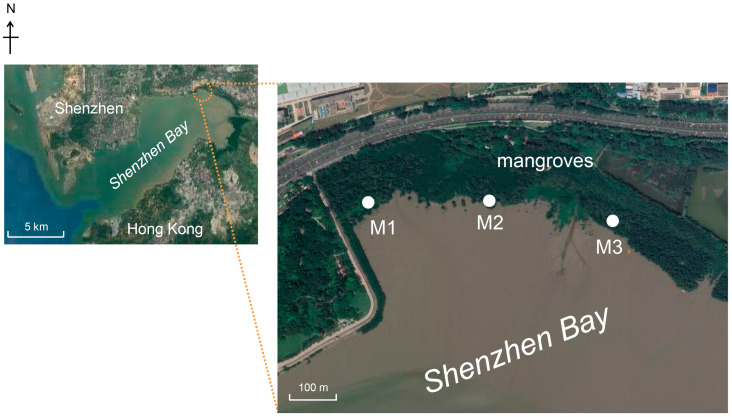
Sampling sites in mangrove sediments of Shenzhen Bay.

**Figure 2 microorganisms-12-01245-f002:**
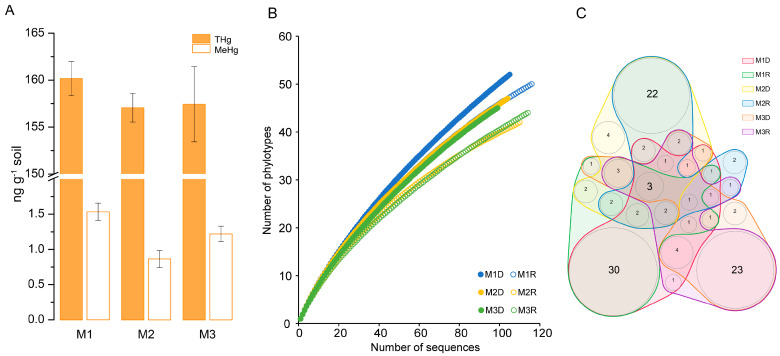
THg and MeHg contents in the sampling sites M1, M2, and M3 of mangrove sediments (**A**); rarefaction curve analysis of the *hgcA* sequences from the gene–based libraries M1D, M2D, and M3D, and transcript–based libraries M1R, M2R, and M3R, based on 80% sequence identity (**B**); Venn analysis of the *hgcA* phylotypes derived from libraries (**C**).

**Figure 3 microorganisms-12-01245-f003:**
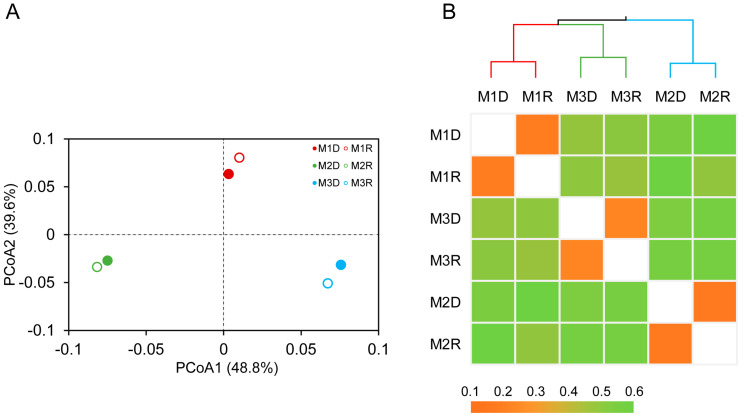
Principal coordinates analysis (PCoA) plot of thetaYC matrix distances for *hgcA* gene (M1D, M2D, and M3D) and transcript (M1R, M2R, and M3R)–harboring communities derived from different sampling sites of mangrove sediments (**A**). Hierarchical clustering heatmap derived from Bray–Curtis matrix distances of *hgcA* genes and transcript–harboring communities (**B**).

**Figure 4 microorganisms-12-01245-f004:**
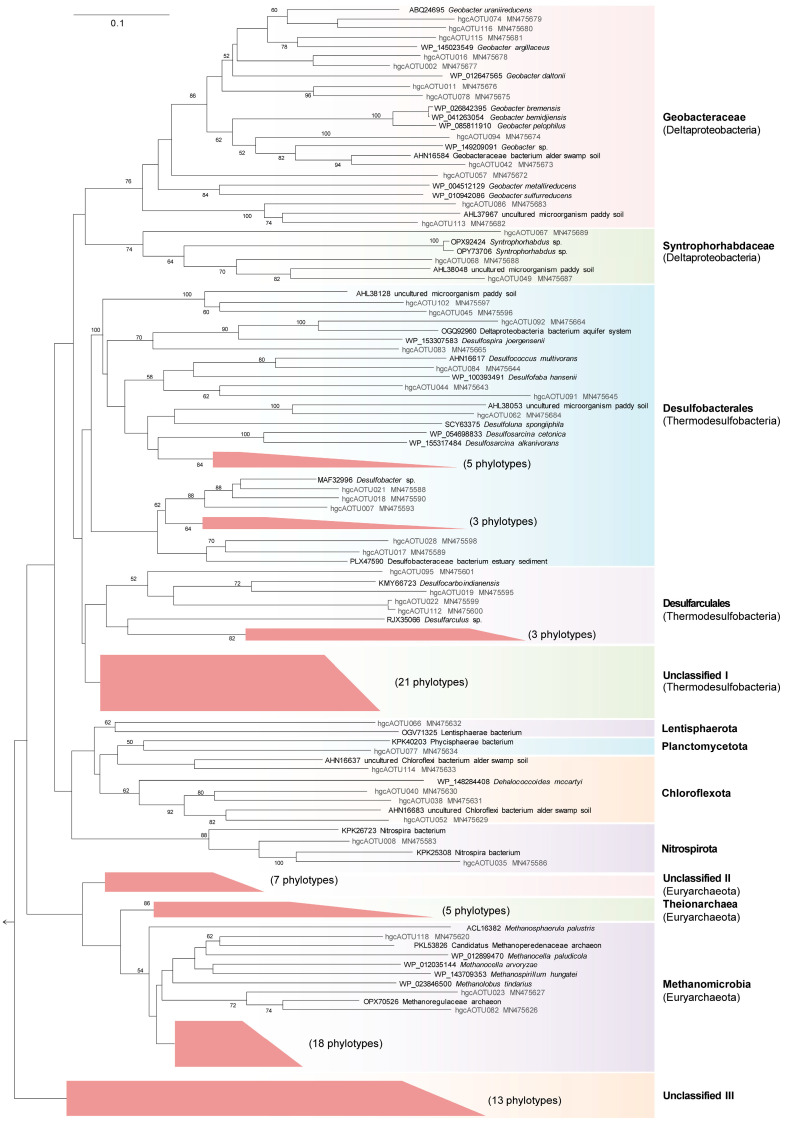
Phylogenetic analysis of the *hgcA* phylotype representatives and their most similar environmental sequences (bold font) identified by BLASTn search in NCBI at the amino acid level. The scale bar represents a 10% sequence divergence per homologous position. The arrow represents the *hgcA* sequence (JCM21531_3779) of a Firmicutes Hg–methylating bacteria, *Hungateiclostridium straminisolvens* JCM 21531. Representatives of the *hgcA* phylotypes were deposited in GenBank under the accession numbers MN475572~MN475689.

**Figure 5 microorganisms-12-01245-f005:**
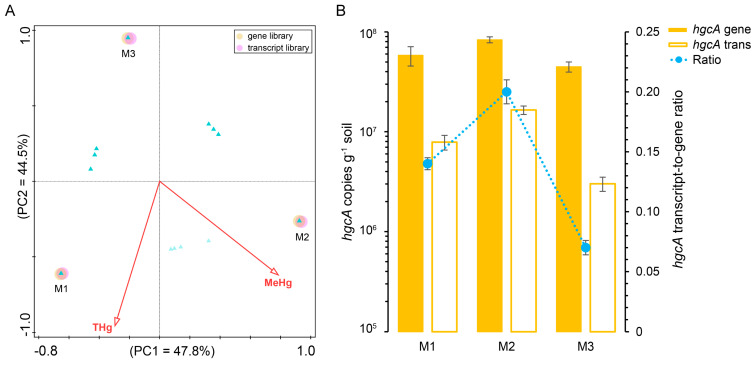
CCA between THg or MeHg content and the structure of *hgcA* genes or transcript library originated from the sampling sites M1, M2, and M3; the triangles represent *hgcA* phylotypes (**A**); quantitative abundance of *hgcA* genes and *hgcA* transcripts and the *hgcA* transcript–to–gene ratio from the sampling sites M1, M2, and M3 of mangrove sediments; the error bar represents the standard deviation (**B**).

**Table 1 microorganisms-12-01245-t001:** Richness estimators and diversity index of *hgcA* sequences from corresponding libraries.

Library	Sobs	Ace	Shannon	InvSimpson	Chao	Coverage
M1D	52	265.0	3.52	26.3	111.5	0.67
M1R	49	82.0	3.33	17.5	72.4	0.77
M2D	47	149.4	3.42	25.6	78.5	0.73
M2R	42	69.4	3.21	15.2	65.1	0.80
M3D	45	183.6	3.29	20.3	76.5	0.72
M3R	44	58.1	3.16	14.7	63.2	0.76

**Table 2 microorganisms-12-01245-t002:** Pearson’s correlation analysis between *hgcA* genes or transcripts and THg or MeHg contents in mangrove sediments.

	*hgcA* Gene ^1^	*hgcA* Trans ^2^	Ratio ^3^	THg ^4^	MeHg ^5^
*hgcA* gene					
*hgcA* trans	0.996 *				
Ratio	0.981	0.984			
THg	0.352	0.364	0.527		
MeHg	0.997 *	0.999 *	0.987	0.382	

^1^ *hgcA* gene, *hgcA* gene abundance; ^2^ *hgcA* trans, *hgcA* transcript abundance; ^3^ ratio, *hgcA* transcript–to–gene ratio; ^4^ THg, THg content; ^5^ MeHg, MeHg content; *, significant correlations *p* < 0.05.

## Data Availability

The data is contained within the article.

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
