# Peer review of "Functional Genes and Transcripts Indicate the Existent and Active Microbial Mercury-Methylating Community in Mangrove Intertidal Sediments of an Urbanized Bay"

_microorganisms, 2024, doi:10.3390/microorganisms12061245_

Round 1

Reviewer 1 Report

Comments and Suggestions for Authors

The manuscript entitled “Functional genes and transcripts indicate the existent and active microbial mercury-methylating community in mangrove intertidal sediments of an urbanized bay» is devoted to an important problem Identification of Hg methyltransferase gene (hgcA) in in in mangrove sediment microbial consortia which can result in the accumulation of the neurotoxic methyl-mercury along the urbanized Shenzhen Bay, China. Authors findings gain better insights of the microbial Hg methylation drivers in mangrove sediments, which would be helpful for understanding the MeHg biotransformation. To my mind this manuscript is topical and corresponding to the aims and scopes of the “Microorganisms” journal. This study combines genetic and transcriptomic and analytical methods of analysis, which makes it possible to establish the relationship between the level of contamination in samples with their phylogenetic composition and functional diversity. Overall, I really liked the article and I am ready to recommend it for publication after correcting several comments.

General and specific Comments.

1. It is worthwhile to provide data on the chemical composition of water samples and the mineral composition of sediment in supplementary. The authors discovered representatives of the sulfur and nitrogen cycles (Geobacteraceae, Syntrophorhabdaceae, Desulfobacterales, Nitrospirae Planctomycetes), it would be important to understand their role in these habitats. It is worth giving the pH value and redox potential of the samples. The behavior of mercury in anoxic zones with a high content of reduced sulfur is not as dangerous as in oxide zones. In this case, this remark can be considered my wish; it does not reduce the high level of work.

2. In my opinion, it is worth expanding the conclusion a little.

3. In the abstract it is worth clarifying the dominant genera in Deltaproteobacteria and Methanomicrobia

Author Response

Reviewer 1

The manuscript entitled “Functional genes and transcripts indicate the existent and active microbial mercury-methylating community in mangrove intertidal sediments of an urbanized bay» is devoted to an important problem Identification of Hg methyltransferase gene (hgcA) in in in mangrove sediment microbial consortia which can result in the accumulation of the neurotoxic methyl-mercury along the urbanized Shenzhen Bay, China. Authors findings gain better insights of the microbial Hg methylation drivers in mangrove sediments, which would be helpful for understanding the MeHg biotransformation. To my mind this manuscript is topical and corresponding to the aims and scopes of the “Microorganisms” journal. This study combines genetic and transcriptomic and analytical methods of analysis, which makes it possible to establish the relationship between the level of contamination in samples with their phylogenetic composition and functional diversity. Overall, I really liked the article and I am ready to recommend it for publication after correcting several comments.

 Author response: Thank you for your comments and suggestions!

General and specific Comments.

  1. It is worthwhile to provide data on the chemical composition of water samples and the mineral composition of sediment in supplementary. The authors discovered representatives of the sulfur and nitrogen cycles (Geobacteraceae, Syntrophorhabdaceae, Desulfobacterales, Nitrospirae Planctomycetes), it would be important to understand their role in these habitats. It is worth giving the pH value and redox potential of the samples. The behavior of mercury in anoxic zones with a high content of reduced sulfur is not as dangerous as in oxide zones. In this case, this remark can be considered my wish; it does not reduce the high level of work.

Author response: Thank you for pointing this out. The mineral composition of sediment and the pH value and redox potential of the samples were added in the Supplementary information document as follows:

Table S1 Physicochemical parameters of the sampling sites in the intertidal sediments of Shenzhen Bay

Parameter

Site1

Site2

Site3

pH

6.67

6.72

6.83

Eh (mV)

-112

-107

-121

TOC (mg g-1)

6.98

7.13

7.4

TN (mg g-1)

0.68

0.75

0.65

The revised text reads were added in the revised manuscript as follows:

L124-L128:

2.2. Physicochemical parameters, THg, and MeHg content analysis

Total organic carbon (TOC), and total nitrogen (TN) contents in sediment samples were measured with a TruMac CNS Macro Analyser (LECO, USA). The pH and redox potential (Eh) values of each sample were determined using a pH/ORP/Temp Portable Meter 6010N (JENCO, USA) prior to collection.

L247-L249:

3.1. Physicochemical characteristics, THg, and MeHg content in mangrove sediments

Physicochemical parameters analysis showed a weakly acidic (pH 6.67 ~ 6.83) and reduced (Eh -121 mV ~ -107 mV) environmental background (Table S1).

  1. In my opinion, it is worth expanding the conclusion a little.

Author response: We agree with the reviewer’s opinion. Accordingly, the sentence “A complex and diverse community was uncovered” was expanded as “A complex and diverse community was uncovered, spanning the Thermodesulfobacteria (mainly Geobacteraceae and Desulfobacterales), Euryarchaeota (mainly Methanomicrobia), Chloroflexota, Nitrospirota, Planctomycetota, and Lentisphaerota taxa.” in the “5. Conclusions” part, L467-470

  1. In the abstract it is worth clarifying the dominant genera in Deltaproteobacteria and Methanomicrobia

Author response: We agree with the reviewer’s opinion. We changed the sentence “The hgcA genes and transcripts associated with Deltaproteobacteria and Methanomicrobia dominated the hgcA-harboring communities,”into “The hgcA genes and transcripts associated with Thermodesulfobacteria [mainly Geobacteraceae, Syntrophorhabdaceae, Desulfobacterales, and Desulfarculales (these four lineages were previously classified into the Deltaproteobacteria taxon)] and Euryarchaeota (mainly Methanomicrobia and Theionarchaea) dominated the hgcA-harboring communities,” in “Abstract” part L20-L22.

Reviewer 2 Report

Comments and Suggestions for Authors

In the Article “Functional Genes and Transcripts Indicate the Existent and Active Microbial Mercury-Methylating Community in Mangrove Intertidal Sediments of an Urbanized Bay”, Guofang Feng and Sanqiang Gong studied the microorganisms potentially involved in mercury (Hg) methylation processes in mangrove sediments through the Hg methyltransferase gene hgcA. Diversity, abundance and expression levels of the functional biomarker gene were analyzed in the context of total Hg and MeHg concentrations in a heavily polluted coastal area, in order to identify the microorganisms potentially driving Hg methylation in mangrove sediments and the environmental factors influencing this transformation. The work presents interesting information regarding these organisms, although further analyses are needed to fully explore the generated data. The main points are the following:

Line 182, What was the rationale for choosing a 20% nucleotide dissimilarity threshold? A 60 % nucleotide similarity threshold was used in the paper cited for the hgcA primer set (ref 15, Bravo et al., 10.1038/s41396-017-0007-7). On the other hand, more recently Gionfriddo et al (ref 56, 10.3389/fmicb.2020.541554) used a 90% sequence identity cut-off. Interestingly, these authors evaluating the use of both nucleotide and deduced amino acid sequences using mock communities.

Line 281, Phylogenetic information of hgcA phylotypes. A more in-depth taxonomic assignment analysis, based on LCA algorithm for instance (Megan6) is needed. Few GenBank sequences are not enough to assess the potential hosts of these genes. For instance, a LCA approach from deduced amino acid sequences was used in Gionfriddo et al (10.3389/fmicb.2020.541554).

Lines 343-365, The qPCR data was informed as copies/g of soil, and the RT-qPCR data as transcript to gene ratios. Were experimental biases considered? For instance, were potential differences in DNA and RNA yield values among samples considered in the analysis of qPCR and RT-qPCR data? It could vary, and affect the results. How the statistical analyses compare when referred as copies/DNA or RNA concentrations?

Line 384, Any hypotheses regarding why MeHg content was a significant factor affecting the hgcA-harboring community structure, in contrast with previous studiees? Could it be related to the toxicity of MeHg? It has been proposed that MeHg is exported from methylating cells, thus reducing toxicity to the methylator (Lin et al., 10.1093/gbe/evad051).

Other comments:

Line 154, 2 μl DNA or cDNA, can you specify ng of DNA or cDNA?

Lines 151-153, please indicate expected length of the amplified products, based on the position of the primers within the gene. What is the coverage of the primer set? It can be checked against an alignment of hgcA genes from an up to date database of genome sequences (IMG/M, for instance)? Primer set was first published 10 years ago, although was modified in a 2018 work. Complete taxa could be missing from the analysis based on the primer coverage. This is in addition to the low sequence coverage values reached in the gene libraries, which is common when using a cloning strategy instead of high-throughput sequencing.

Figure 4, font is too small

Line 243, THg instead of Hg. Only briefly at the end of the discussion, the authors mention the different forms of Hg and their bioavailability, and none related to the presented data.

Comments on the Quality of English Language

Good overall quality of English, a few expressions could be rephrased for clarity:

Line 33, from terrene to ocean

Line 75, conducted to predicate the

Line 80, the environmental fluctuating

Lines 358-359, 2). Whereas positive

Line 429, may functionate

Line 438, This might because 

Line 449, The structure traits

Besides,

Author Response

Reviewer 2 In the Article “Functional Genes and Transcripts Indicate the Existent and Active Microbial Mercury-Methylating Community in Mangrove Intertidal Sediments of an Urbanized Bay”, Guofang Feng and Sanqiang Gong studied the microorganisms potentially involved in mercury (Hg) methylation processes in mangrove sediments through the Hg methyltransferase gene hgcA. Diversity, abundance and expression levels of the functional biomarker gene were analyzed in the context of total Hg and MeHg concentrations in a heavily polluted coastal area, in order to identify the microorganisms potentially driving Hg methylation in mangrove sediments and the environmental factors influencing this transformation. The work presents interesting information regarding these organisms, although further analyses are needed to fully explore the generated data.

Author response: Thank you for your comments and suggestions!

The main points are the following:

  1. Line 182, What was the rationale for choosing a 20% nucleotide dissimilarity threshold? A 60 % nucleotide similarity threshold was used in the paper cited for the hgcA primer set (ref 15, Bravo et al., 10.1038/s41396-017-0007-7). On the other hand, more recently Gionfriddo et al (ref 56, 10.3389/fmicb.2020.541554) used a 90% sequence identity cut-off. Interestingly, these authors evaluating the use of both nucleotide and deduced amino acid sequences using mock communities.

Author response: Thank you for pointing this out. We think this is an excellent suggestion. The 20% nucleotide dissimilarity threshold for hgcA phylotype referred to the research “Liu et al. Analysis of the microbial community structure by monitoring an Hg methylation gene (hgcA) in paddy soils along an Hg gradient. Appl Environ Microbiol. 2014 May;80(9):2874-9.”, as the ref [12] in this manuscript. Following the reviewer’s suggestion, we will carefully choose more representative and more rigorous similarity thresholds of hgcA gene in our further investigations.

  1. Line 281, Phylogenetic information of hgcA phylotypes. A more in-depth taxonomic assignment analysis, based on LCA algorithm for instance (Megan6) is needed. Few GenBank sequences are not enough to assess the potential hosts of these genes. For instance, a LCA approach from deduced amino acid sequences was used in Gionfriddo et al (10.3389/fmicb.2020.541554).

Author response: Thank you for pointing this out. We agree with the reviewer’s opinion. Accordingly, taxonomic assignment analysis of hgcA phylotypes were performed based on LCA algorithm, and the phylogenetic tree was re-edited as the new Figure 4. The description was added in L204-L206: “Taxonomic classifications can be assigned with the lowest common ancestor (LCA) of the subtree where the sequence was placed by the ‘guppy classify’ command [41] with a classification cutoff of 80%.”, and corresponding reference was added as ref [41].

  1. Lines 343-365, The qPCR data was informed as copies/g of soil, and the RT-qPCR data as transcript to gene ratios. Were experimental biases considered? For instance, were potential differences in DNA and RNA yield values among samples considered in the analysis of qPCR and RT-qPCR data? It could vary, and affect the results. How the statistical analyses compare when referred as copies/DNA or RNA concentrations?

Author response: Thank you for pointing this out. we appreciate the reviewer’s comments.

(1) A previous study to investigate the effects of different DNA/RNA extract method and RNA treated method showed that, the DNA elution method is recommended for evaluating the total diversity of microorganisms in marine sediments, and RNA can be used after purification to investigate active communities[1]. Referring to this methodology, firstly, the DNA and RNA were extracted using the Kits using the DNA/RNA elution methods, and have been verified to extract high quality DNA and RNA in our lab. Secondly, the RNase-free DNase I was used to digest the potential residual DNA in RNA, and after that the RNA was purified referring to the methodology reported above. Thirdly, the prurified RNA was as the negative control during qPCR perform. Therefore, we have taken reasonable measures to minimize the experimental biases during DNA/RNA extraction step.

(2) In our study, quantitative PCR procedures of hgcA gene and transcript referred to the previous qPCR study of hgcA transcript (ref 42) to minimize the potential experimental biases during qPCR step.

(3) The abundance of hgcA gene and hgcA transcript were both expressed in the copies/ g soil. In detail, after extraction, the DNA and RNA contents were expressed as ng DNA / g soil and ng RNA / g soil, respectively. The quantitative PCR data were expressed as copies of hgcA gene/ ng DNA, and copies of hgcA transcript/ ng RNA, which can be transformed to copies of hgcA gene/ g soil, and copies of hgcA transcript/ g soil, respectively. Then, the abundance of hgcA gene and hgcA transcript can be compared in the form of “transcript-to-gene ratio”. Similar conditions are performed for the DNA and RNA level revelation and “transcript-to-gene ratio” comparison of various genes, i.e., amoA,nirS, nirK, 16S rRNA anammox from various sedimentary environments[2],[3],[4].

  1. Line 384, Any hypotheses regarding why MeHg content was a significant factor affecting the hgcA-harboring community structure, in contrast with previous studies? Could it be related to the toxicity of MeHg? It has been proposed that MeHg is exported from methylating cells, thus reducing toxicity to the methylator (Lin et al., 10.1093/gbe/evad051).

Author response: Thank you for pointing this out. We agree with the reviewer’s assessment. Previous studies indicated that, the diversity of the observed hgcA gene could be associated with the MeHg content in paddy soils[5]. Besides, hgcA gene abundance was significantly correlated with MeHg content in Hg-polluted paddy soils[6], Hg mining area paddy soils[7], and the fluctuating zone sediments[8]. As the reviewer’s suggestion, MeHg content as a significant factor affecting the hgcA-harboring community structure could be related to the toxicity of MeHg, since it has been proposed that MeHg is exported from methylating cells to reduce the toxicity to the methylator. Therefore, we revised the sentences in L399-L402: “It has been proposed that MeHg could be exported from the metabolically methylating cells to reduce the toxicity to the microbial methylators [52]. Such a finding was similar to that in paddy soils anthropogenically treated with different mercury species [53]”, and corresponding references were added as ref [52] and [53].

Other comments:

  1. Line 154, 2 μl DNA or cDNA, can you specify ng of DNA or cDNA?

Author response: Thank you for pointing this out. We agree with the reviewer’s suggestion. This sentence was described as “PCR was conducted in a total volume of 40 μl containing 10 ng DNA or 2 μl cDNA (equivalent to 4 ng RNA),” in L165. Specially, since cDNA concentration is difficult to measure accurately and directly, we converted it to RNA concentration to express it.

  1. Lines 151-153, please indicate expected length of the amplified products, based on the position of the primers within the gene. What is the coverage of the primer set? It can be checked against an alignment of hgcA genes from an up-to-date database of genome sequences (IMG/M, for instance)? Primer set was first published 10 years ago, although was modified in a 2018 work. Complete taxa could be missing from the analysis based on the primer coverage. This is in addition to the low sequence coverage values reached in the gene libraries, which is common when using a cloning strategy instead of high-throughput sequencing.

Author response: Thank you for pointing this out. We appreciate the reviewer’s suggestion and comments.

(1) The length of the amplified products and the position of the primers within the gene were added as “which located the 249 ~ 269 and 877 ~ 903 regions of the hgcA gene (B2D07_07835) from the methylating species Desulfococcus multivorans DSM 2059. The forward primer hgcA261F  targets the highly conserved putative cap helix, while the reverse primer hgcA912R  targets a less conserved motif between the third and fourth predicted transmembrane helices near the C-terminus, producing an amplicon of ~ 650 bp [8] ” in L159-L164.

  1. As the reviewer’s comments, the new primer set (i.e., ORNL-HgcAB-uni-F and ORNL-HgcAB-uni-32R) for hgcA and hgcB could cover a broad-range of putative methylators in environments than the early reported primer set (hgcA261F and hgcA912R in our study). However, our research focused on both DNA and RNA level revelation of hcgA sequences, the new primer set for hgcA and hgcB is not suitable for RNA level research, since hgcA and hgcB are two separate transcripts after transcription in cells, which cannot be amplified using the new primer set covering both hgcA and hgcB. A former study showed that primer set primer set for hgcA and hgcB cannot detect their transcripts at RNA level[9]. Therefore, we still using the early reported hgcA261F and hgcA912R primer set in this study. Of course, according to the reviewer’s suggestion, our further DNA level investigation would carefully choose the broad-range primer set for hgcA gene.
  2. Figure 4, font is too small

Author response: We agree with the reviewer’s comments. Figure 4A was revised after LCA based taxonomic assignment of hgcA phylotypes as the new Figure 4, and the Figure 4B showing the relative abundance of each phylotype in corresponding clone library was moved to the Supplementary information document as Figure S1. After the above operations, the font is enlarged.

  1. Line 243, THg instead of Hg. Only briefly at the end of the discussion, the authors mention the different forms of Hg and their bioavailability, and none related to the presented data.

Author response: Thank you for pointing this out. THg instead of Hg was performed in the revised manuscript L257 and Figure 2A. We appreciate the reviewer’s comments. A previous study has verified that different Hg species would affect the community structure of hgcA-harboring community[10]. Thus, different forms of Hg and their bioavailability associated with hgcA-harboring community would be an important research direction which will be considered as in our further investigation.

.  Comments on the Quality of English Language

Good overall quality of English, a few expressions could be rephrased for clarity:

Line 33, from terrene to ocean

Author response: changed into “from terrestrial to marine” in L35.

Line 75, conducted to predicate the

Author response: changed into “performed to predict” in L77.

Line 80, the environmental fluctuating

Author response: changed into “the environmental fluctuation” in L82.

Lines 358-359, 2). Whereas positive

Author response: changed into “Whereas a positive” in L373.

Line 429, may functionate

Author response: changed into “drive” in L446.

Line 438, This might because

Author response: changed into “This could be because” in L455.

Line 449, The structure traits

Author response: changed into “The structure characteristics” in L466.

[1] Li et al. 2022. Effects of DNA and RNA extraction methods for the evaluation of ciliate diversity in marine sediments. Marina Sciences, 46(7): 52-60.

[2] Marshall AJ, et al. Temporal profiling resolves the drivers of microbial nitrogen cycling variability in coastal sediments. Sci Total Environ. 2023, 856(Pt 1):159057.

[3] Hu J, et al. Soil health management enhances microbial nitrogen cycling capacity and activity. mSphere. 2021, 6:10.

[4] Bodle KB, et al. Treatment performance and microbial community structure in an aerobic granular sludge sequencing batch reactor amended with diclofenac, erythromycin, and gemfibrozil. Front Microbiomes. 2023, 2:1242895.

[5] Liu C et al. Geochemical mercury pools regulate diverse communities of hgcA microbes and MeHg levels in paddy soils. Environ Pollut 2023, 334, 122172.

[6] Liu, Y et al. Analysis of the microbial community structure by monitoring an hg methylation gene hgcA in paddy soils along an Hg gradient. Appl Environ Microbiol 2014, 80, 2874-2879.

[7] Liu, X. et al. Diversity of microbial communities potentially involved in mercury methylation in rice paddies surrounding typical mercury mining areas in China. MicrobiologyOpen 2018, 7, e00577.

[8] Du, H et al. Mercury-methylating genes dsrB and hgcA in soils/sediments of the Three Gorges Reservoir. Environ Sci Pollut Res Int 2017, 24, 5001-5011.

[9] Bae HS, et al. Syntrophs dominate sequences associated with the mercury methylation-related gene hgcA in the water conservation areas of the Florida Everglades. Appl Environ Microbiol. 2014, 80(20):6517-26.

[10] Liu C, et al. Geochemical mercury pools regulate diverse communities of hgcA microbes and MeHg levels in paddy soils. Environ Pollut. 2023, 334:122172.